# Improving citizen-government interactions with generative artificial intelligence: Novel human-computer interaction strategies for policy understanding through large language models

**Lixin Yun[1], Sheng Yun[2]\*, Haoran Xue[2]**

**1** School of Humanities and Social Sciences, Qingdao Agricultural University, Qingdao, Shandong, China,
**2** Computer and Information Science, Fordham University, New York, New York, United States of America

\* syun13@fordham.edu

## Abstract

Effective communication of government policies to citizens is crucial for transparency and engagement, yet challenges such as accessibility, complexity, and resource constraints obstruct this process. In the digital transformation and Generative AI era, integrating Generative AI and artificial intelligence technologies into public administration has significantly enhanced government governance, promoting dynamic interaction between public authorities and citizens. This paper proposes a system leveraging the Retrieval-Augmented Generation (RAG) technology combined with Large Language Models (LLMs) to improve policy communication. Addressing challenges of accessibility, complexity, and engagement in traditional dissemination methods, our system uses LLMs and a sophisticated retrieval mechanism to generate accurate, comprehensible responses to citizen queries about policies. This novel integration of RAG and LLMs for policy communication represents a significant advancement over traditional methods, offering unprecedented accuracy and accessibility. We experimented with our system with a diverse dataset of policy documents from both Chinese and US regional governments, comprising over 200 documents across various policy topics. Our system demonstrated high accuracy, averaging 85.58% for Chinese and 90.67% for US policies. Evaluation metrics included accuracy, comprehensibility, and public engagement, measured against expert human responses and baseline comparisons. The system effectively boosted public engagement, with case studies highlighting its impact on transparency and citizen interaction. These results indicate the system's efficacy in making policy information more accessible and understandable, thus enhancing public engagement. This innovative approach aims to build a more informed and participatory democratic process by improving communication between governments and citizens.

**Data Availability Statement:** All relevant data for this study are publicly available from the Harvard Dataverse repository (DOI: 10.7910/DVN/7PSN0L).

**Funding:** The author(s) received no specific funding for this work.

**Competing interests:** The authors have declared that no competing interests exist.

# 1 Introduction

Communicating government policies to citizens is a fundamental aspect of public administration and governance, serving as a bridge between the decision-makers and the populace whose lives are directly impacted by these policies and plans [1]. Historically, this communication has taken various forms, ranging from official documents, press releases, and public announcements to more interactive platforms such as public forums, consultations, and, more recently, digital and social media channels [2]. These methods aim to ensure transparency, promote public awareness, and stimulate citizen engagement with governmental processes. However, despite these efforts, numerous challenges persist, including the accessibility of information, the complexity of policy language, and the general public's understanding of governmental procedures [3]. For instance, detailed policy documents and press releases may ensure broad dissemination but often fail to engage citizens effectively due to their complexity and the bureaucratic language used, leaving a significant portion of the population disengaged or misinformed about policies that have a substantial impact on their lives [4, 5]. Disengagement can cause the citizens to be indifferent to politics and society, and misinformation may lead to trust issues in government policies which can be dangerous in a crisis.

The challenges are further compounded by financial and logistical constraints faced by government departments dedicated to public communication. The significant costs associated with both traditional and digital dissemination methods, combined with the slow response times to citizen inquiries and insufficient manpower [3, 5, 6], especially in financially constrained areas, aggravates the situation. Moreover, the often limited understanding of complex policies by the staff tasked with their communication can lead to misinformation, further eroding public trust in government institutions. These issues show a critical gap in the effectiveness of existing communication strategies, necessitating innovative solutions to bridge the information divide between governments and their citizens.

Artificial Intelligence (AI) and Large Language Models (LLMs) have emerged as powerful tools capable of transforming various sectors, including public administration and governance [7]. AI refers to machines designed to mimic human intelligence, performing tasks such as generalizing complex texts, problem-solving, and decision-making, while LLMs are a subset of AI, focusing on understanding and generating human language. These technologies have demonstrated remarkable capabilities, including the ability to process and analyze vast amounts of data, understand complex patterns, and generate coherent, informative content [8–10]. Their application spans numerous fields, offering potential solutions to longstanding challenges by enhancing efficiency, accuracy, and accessibility [7, 9]. In public administration, the integration of AI and LLMs has the potential to revolutionize the communication of government policies, making them more accessible and comprehensible to the general public.

This convergence of AI and LLMs in public administration not only smooths information processing but also revolutionizes how government policies are communicated to and understood by the public [11]. Their capacity to digest and distill vast amounts of complex documents into concise, understandable summaries holds the potential to democratize access to policy information, making it significantly more accessible to the general public [8]. Furthermore, LLMs facilitate real-time, interactive communication, allowing citizens to receive personalized responses to their inquiries. This shift from traditional, unidirectional communication methods towards a more interactive, responsive, and citizen-focused approach promises to enhance public engagement with governmental processes and ensure that policy information is not just available but also comprehensible to the average citizen.

To capitalize on the capabilities of AI and LLMs in enhancing the communication of government policies, various innovative approaches have been explored. The development of

chatbots and virtual assistants, for instance, provides real-time responses to citizen inquiries, embodying the move towards instant, accessible communication [12]. AI-driven platforms that simplify and summarize policy documents into digestible formats are becoming increasingly prevalent, along with the integration of LLMs into government websites and social media channels to boost accessibility and engagement [13, 14]. Furthermore, initiatives like AI-curated newsfeeds on policy developments and virtual town halls moderated by AI for more inclusive public discussions are being tested [14, 15]. These solutions aim to leverage the power of AI directly to transcend the limitations of conventional communication methods, ensuring policy information is not only more accessible but also more comprehensible.

The direct application of LLMs in communicating government policies is fraught with challenges [16]. The complexity and sensitivity of governmental information demand a degree of accuracy, relevance, and security that raw LLM outputs may not consistently provide. The potential for misinformation, biases inherent in the training data, and the ethical implications of AI-generated content pose significant hurdles. These challenges underscore the necessity for a better approach that can harness the strengths of LLMs while addressing their limitations.

To enhance AI's potential in government governance, we propose a system that combines Retrieval-Augmented Generation (RAG) with LLMs. This creates an interactive policy interpreter that combines LLMs' generative capabilities with the precision of retrieval-based systems for more accurate, reliable outputs [17]. This approach reduces bias, unpredictability, and the need for costly customizations, offering a cost-effective solution to improve governance. We tested our application with policy documents in Simplified Chinese and English, showing high reliability, accuracy, and efficiency. This method enhances public service responsiveness and incorporates transparency, accountability, and inclusivity, promising for future government technology research and application.

In this paper, we provide a review of related work in Section 2, highlighting the existing approaches and their limitations. In section 3, we describe the proposed system in detail, including the architecture and implementation of the RAG technology combined with LLMs. We discuss the dataset used for testing and the preprocessing steps involved in Section 4. Section 5 presents the experimental setup, results, and performance analysis, including a comparative analysis with previous studies. And in section 6, we discuss the challenges, limitations, and future directions for this research. Finally, Section 7 concludes the paper with a summary of findings and implications for policy communication.

## 2 Related work

Recent research has highlighted the potential of artificial intelligence (AI) and large language models (LLMs) in enhancing government-citizen communication. Mamalis et al. [11] demonstrated how ChatGPT could revolutionize open government data portals, improving the accessibility and usability of statistical information. Similarly, Androutsopoulou et al. [12] explored AI-guided chatbots for transforming citizen-government communication, emphasizing their potential to provide personalized and efficient public services. Recent studies have also examined the adoption and implementation of AI chatbots in public organizations. Chen et al. [18] investigated the determinants facilitating or impeding chatbot adoption and implementation across U.S. state agencies, identifying factors like ease of use, leadership, and external shocks as key drivers. Wang et al. [19] explored chatbot adoption in Chinese local governments, finding that pressure factors and readiness factors play different roles in the initial adoption versus post-adoption performance stages. Cortés-Cediel et al. [20] conducted a comprehensive analysis of e-government chatbots, identifying trends and challenges in both research literature and

deployed applications, and proposing novel chatbots for exploring open government data and citizen participation content. These studies, however, do not utilize Retrieval-Augmented Generation (RAG) or guarantee the queries from the clients will be answered in the way the authorities expected, potentially limiting their accuracy and adaptability in handling complex, evolving policy information. The application of RAG in policy communication represents a novel approach. Lewis et al. [17] introduced RAG for knowledge-intensive NLP tasks, showing its effectiveness in combining the strengths of retrieval and generation models. This technique has since been adapted for various domains, including public administration. Challenges in implementing AI for government communication have also been identified. Hadi et al. [16] surveyed large language models, highlighting limitations such as potential biases and the need for continuous updates to maintain accuracy, which implies the necessity of integrating the RAG into the system. Kasneci et al. [7] discussed both opportunities and challenges of using LLMs in educational contexts, which can be extended to policy communication. Our proposed approach builds upon these advancements by integrating RAG technology with Langchain, specifically tailored for policy communication. This method addresses the identified challenges by ensuring up-to-date, accurate information retrieval while leveraging the natural language generation capabilities of LLMs.

## 3 Communicating government policies to citizens

### Overview of current communication strategies

Understanding the intricacies of government policies and plans remains a critical challenge for citizens across the globe. In many countries [2], including China, the United States, the United Kingdom, Canada, and Australia, governments have traditionally relied on a combination of public announcements, official documents, websites, and direct citizen engagement through public forums and consultations to disseminate this information, as shown in Fig 1. However, the effectiveness of these methods varies significantly, influenced by factors such as accessibility, the complexity of information presented, and the public's general awareness and understanding of governmental procedures. For instance, China's approach often involves detailed policy documents and press releases on government websites, complemented by media coverage. While this ensures a broad dissemination of information, the complexity of language and the volume of data can overwhelm citizens, hindering their understanding. A more personalized approach that enables the citizen to know what the policies that may largely impact their life is still not very effective and most of the citizens are either not eager or not able to acquire the policies that matter the most.

### Financial and logistical constraints

Like many services in the government, the financial and logistical aspects of these dissemination methods present substantial burdens [21]. Departments dedicated to public communication and engagement often face budgetary constraints, limiting their ability to reach out effectively to all citizens [21]. In the US and Canada, for instance, extensive use of digital platforms and social media has helped in reducing some costs but has not eliminated the need for traditional, more resource-intensive methods such as public meetings and printed materials. These strategies entail significant extra expenses, from the production of materials to the organization of events, straining the already tight budgets of government departments.

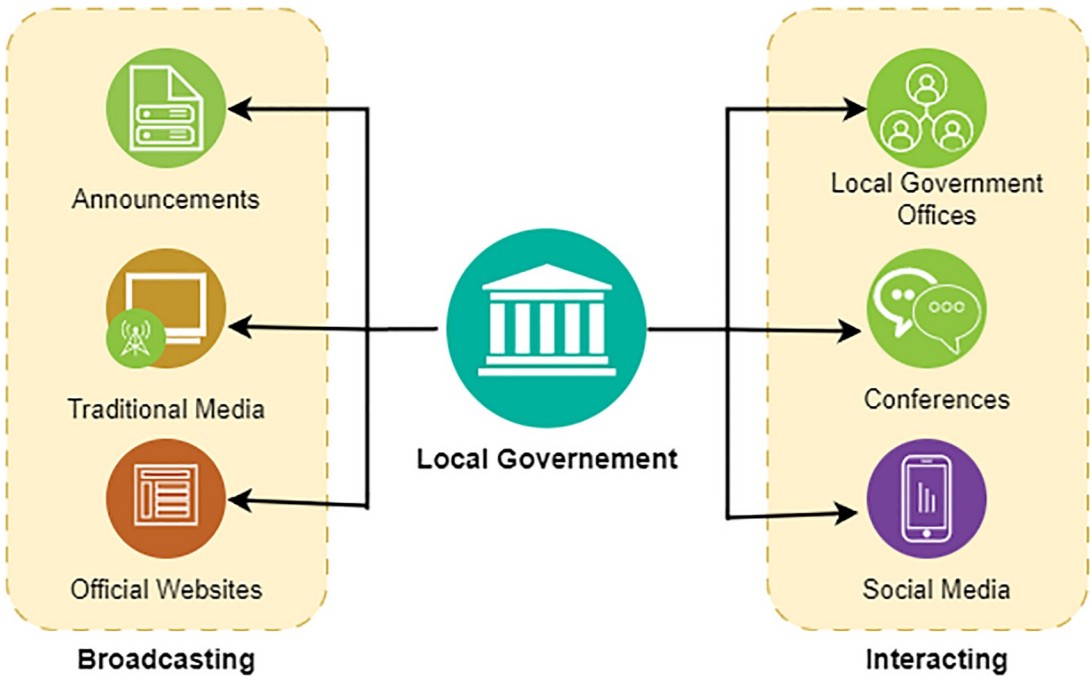

**Fig 1. Current major media of communicating government policies to citizens.**

### Inherent challenges in policy communication

The challenges are exacerbated when considering the responsiveness of government agencies to citizen inquiries [5]. Slow responses are a common issue, partly due to the sheer volume of queries and the bureaucratic processes involved in addressing them. This delay in communication can lead to frustration and dissatisfaction among the public, eroding trust in government institutions. In many cases, the bureaucratic nature of government operations can delay the dissemination of information, resulting in outdated or irrelevant information being provided to the public. This is particularly problematic in emergencies, such as during health crises or natural disasters, where timely and accurate information is critical [22].

Moreover, the cost of manpower is a critical concern. The costs associated with these dissemination activities are substantial and encompass not only the direct expenses of creating and distributing the information but also the operational costs of the departments responsible for these tasks. In countries like the United States and Canada, significant investments are made in digital infrastructure to support the online presence of government agencies, including websites and social media engagement [23, 24]. Similarly, the United Kingdom and Australia have dedicated teams within government departments tasked with public relations and communication, ensuring that policy explanations are clear, accessible, and reach a wide audience. However, these efforts represent a considerable portion of government budgets, with ongoing debates about their cost-effectiveness and impact on public awareness and understanding of government actions.

This problem is particularly acute in rural areas of countries like China, where limited financial resources and the logistical challenges of reaching a dispersed population hinder the government's ability to hire sufficient personnel for effective communication. The result is a gap in the timely provision of information to citizens, who often seek clarity on policies and regulations that directly impact their lives. This illustrates a scenario where the government

struggles to allocate enough human resources to engage with the public effectively. Hiring individuals to bridge the information gap is not always viable, leading to an overreliance on less interactive digital communication platforms. This approach, while cost-effective, fails to address the nuances of personal interaction and can leave the more technologically disadvantaged populations further behind.

Another layer of complexity is added by the staff's limited understanding of policies and plans. Even when personnel is available, their ability to interpret complex information in an accessible manner is not guaranteed, and citizens who would like to inquire about specific policies might be misled by the government staff's lack of knowledge of the new policies. The intricacies of policy and legislative language, coupled with the often intricate implications of these documents, demand a high level of expertise. Unfortunately, the requisite training and knowledge development among staff are areas that frequently suffer due to financial and time constraints [25]. The misunderstanding caused by the ambiguity or mistake made by the staff will not only compromise the effectiveness of the government policy deliveries but also harm the public's trust in the government. This issue is not unique to any single country; it reflects a global challenge within government communication efforts.

Case studies from across the mentioned countries show attempts to mitigate these challenges through innovative solutions. For example, some local governments in China have begun to leverage social media and messaging apps to provide more direct and interactive communication channels. These efforts, however, are but a drop in the ocean of necessary reform. The overarching issue remains: without comprehensive strategies that address the financial, logistical, and educational barriers to effective government communication, citizens remain at a disadvantage in understanding and engaging with the policies that shape their lives. Similarly, in the US, the rollout of healthcare policies has often been met with public confusion and skepticism, in part due to communication missteps and the complexity of the information being conveyed.

Overall, The task of explaining government policies and plans to citizens is fraught with significant challenges. From the financial costs associated with traditional and digital dissemination methods to the slow response times and the insufficient manpower, especially in financially constrained areas, these obstacles are compounded by the staff's often limited understanding of complex policies. As governments seek to navigate these hurdles, the search for more efficient, accessible, and cost-effective communication methods continues, highlighting the critical need for innovation in public administration and engagement strategies.

# 4 Our approach to enhancing government-citizen communication

## Our system using RAG and LLMs

Our approach to improving the communication of government policies to citizens is centered around the innovative application of Retrieval-Augmented Generation (RAG) technology integrated with Langchain, as shown in Fig 2. We employ GPT-3.5-turbo for question answering, with documents converted to manageable formats and segmented for efficient retrieval. The RAG technology enables the system to dynamically retrieve relevant documents from a comprehensive knowledge base and generate responses that are both accurate and contextually relevant. By leveraging the capabilities of large language models (LLMs) and combining them with a sophisticated retrieval mechanism, our system is designed to deliver accurate, up-to-date, and comprehensible information on government policies directly to the public. Our system not only generates responses based on a vast corpus of governmental documents but also ensures that the information provided is specifically tailored to the user's query, making complex policy information accessible and understandable to the layperson. The key innovation

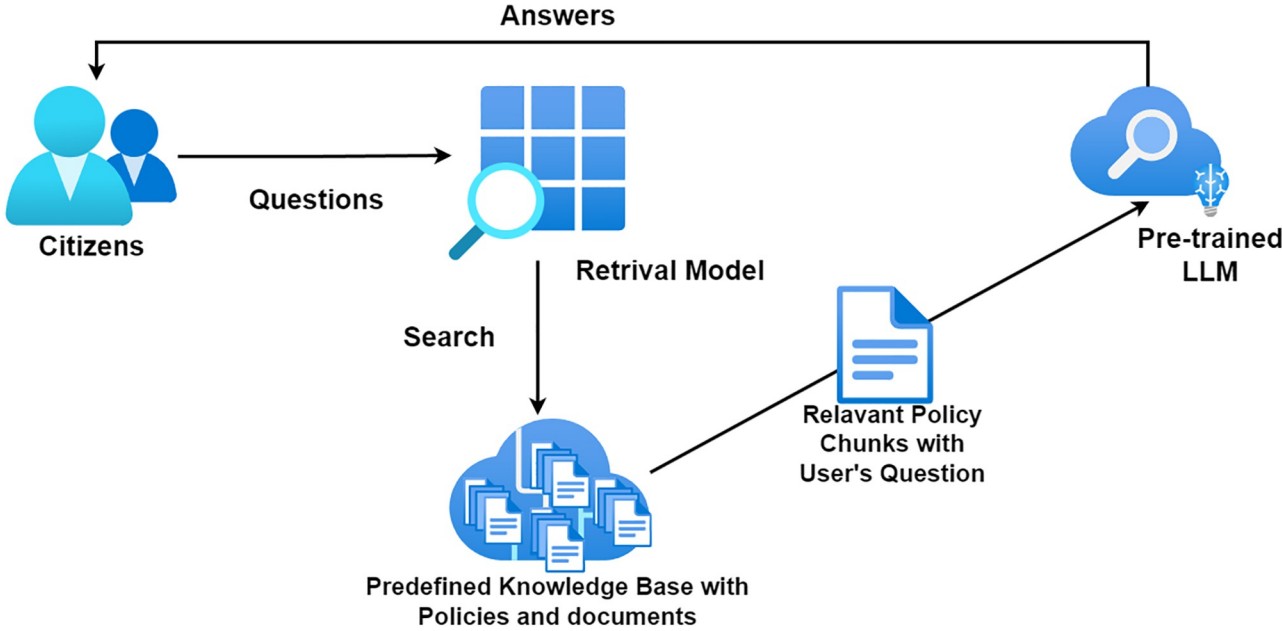

**Fig 2. Retrieval augmented generation with llm for communication of government policy to citizen.**

here is the system's ability to dynamically retrieve relevant documents and synthesize this information into clear, concise responses using LLM's ability to generalize the ideas only based on the retrieved information, We believe this can bridge the gap between the intricate world of policy-making and the everyday lives of citizens.

To further enhance the efficacy of our approach, we employ Langchain to facilitate the seamless integration of the retrieval and generation processes. Langchain's role is important, as it acts as the backbone of our system, enabling the dynamic updating of the knowledge base with the latest policies and regulations. This ensures that our system remains current and relevant, capable of addressing the public's questions with the most recent information, and it can help the retrieval component to identify the most relevant information regarding the query. We then can ensure that the responses are not just accurate but also engaging and up-to-date for the general public.

## Implementation in citizen communication

To maximize the impact of our approach, it is crucial to ensure broad and easy access for all citizens. This can be achieved by integrating our RAG-based system within existing government digital platforms, such as official websites, mobile apps chat-bots, and social media channels. By embedding the system in platforms that citizens already use to seek information, we can significantly enhance the accessibility of complex policy information. Furthermore, the implementation should be spearheaded by departments that are directly involved in public communication and citizen engagement, such as the Department of Public Information or equivalent entities within local and national government structures. These departments, with their expertise in communication strategies and their existing infrastructure, are best positioned to manage the integration of the RAG system, ensuring that it is effectively utilized to improve the dialogue between the government and its citizens.

To illustrate the real-world applicability of our system, consider a scenario where a government agency implements our RAG-based system to communicate a new environmental policy.

The system is integrated into the agency's website and mobile app, providing citizens with instant access to policy details. A citizen queries about specific regulations regarding waste management, and the system retrieves relevant documents, synthesizing a clear and concise response. This approach not only enhances transparency but also ensures that citizens receive accurate information, thereby improving public understanding and compliance with the policy.

In addition to digital integration, physical access points such as public libraries, community centres, and government offices can be equipped with terminals or kiosks where citizens without internet access at home can interact with the system. These digital and physical access points ensure that the benefits of the RAG system are extended to the widest possible audience, including those in rural or underserved areas. Training for staff members at these access points will also be necessary to assist citizens in navigating the system and understanding the responses generated, thereby enhancing the overall effectiveness of the communication strategy.

## Building the knowledge base for RAG

The foundation of our RAG system is its knowledge base, which must be comprehensive, authoritative, and up-to-date to ensure the accuracy and relevance of the information provided to citizens. To construct this knowledge base, we will compile a wide array of government documents, policies, regulations, and related materials. This compilation process involves close collaboration with various government departments and agencies to secure access to current and historical documents. Additionally, our knowledge base must be incorporated with a process that can have continuous updates, with a mechanism in place for the timely incorporation of new policies, amendments, and relevant announcements into the knowledge base. This dynamic updating process is critical for maintaining the system's effectiveness in providing accurate and current information.

Beyond the initial compilation, the management of the knowledge base requires the application of advanced categorization and indexing techniques. These techniques ensure that the retrieval component of the RAG system can efficiently locate and extract the most relevant documents in response to a user's query. The use of metadata, such as document type, publication date, and policy area, will facilitate this process, enabling more precise matching and thereby enhancing the overall quality of the information retrieved. This meticulous organization of the knowledge base is essential for the system's ability to deliver targeted and coherent responses, reflecting the complexities and nuances of government policies in a manner that is accessible to the general public.

## Potential improvement on government-citizen communication

Our approach represents a significant leap forward in the way government policies are communicated to citizens. By leveraging the RAG system integrated with Langchain, we can transform the traditionally complex and opaque process of policy dissemination into a user-friendly, engaging, and interactive experience. This system not only makes policy information more accessible but also more understandable, empowering citizens to gain insights into government actions that directly affect their lives. The real-time, personalized responses provided by the system ensure that citizens can receive specific information relevant to their needs and concerns, thereby fostering a more informed and engaged public. Ultimately, our approach enhances transparency, builds trust, and encourages greater citizen participation in governance processes, contributing to the development of a more responsive and accountable government.

## 5 Details in the RAG component in our approach

### Overview of our RAG component

At the core of our system is Retrieval-Augmented Generation (RAG) utilizing Langchain technology. RAG, at its essence, is designed to harness the vast knowledge embedded within large language models (LLMs) while integrating a retrieval mechanism that pulls in specific, relevant information from a designated knowledge base (database or corpus of documents). This integration ensures that the responses generated are not only linguistically coherent but also deeply grounded in the most current and accurate data available. Our adoption of RAG through Langchain is tailored specifically to address the unique challenges faced in public administration, particularly in the dissemination of complex policy information.

### Architecture of our RAG component

The architecture of our Retrieval-Augmented Generation (RAG) model is designed to optimize the communication of complex government policies to the public with unparalleled precision and accessibility. At the heart of this architecture lies the document retriever component, a sophisticated algorithm designed to navigate the vast expanse of government documentation using a semantic search based on the question that it has been asked. This component leverages advanced search techniques and natural language processing (NLP) capabilities to sift through the knowledge base, identifying documents that most closely match the user's query. By prioritizing relevance and recency, the retriever ensures that the foundational information for each response is pertinent and reflects the latest legislative and policy updates. This semantic search process is critical, as it guarantees that the system's outputs are grounded in the most current and relevant facts, a prerequisite for accurate and trustworthy communication.

Before the retrieval of documents in our knowledge base, we need to build our knowledge base. To achieve that, we use a document reader to load vast amounts of documents to compile a knowledge base that can be used by the retriever. This element of the system is tasked with the complex job of parsing and understanding the content within the selected documents. Employing multi-modal NLP techniques, the document reader analyses and embeds (encodes) the text, tables, and images, identifying key facts, arguments, and nuances contained within. It is capable of discerning the relevance of different pieces of information, categorizing them by significance later to the user's query.

The final stage in our RAG model's architecture involves the generative component, which is where we use a Large Language model to answer the question from the retrieved information. What sets this stage apart is LLM's ability to tailor the complexity and style of the response to suit the layperson's understanding, thereby demystifying government policies and making them accessible to a broader audience, while maintaining credibility by restricting the LLM to only answer the question based on the information retrieved from our pre-defined knowledge base. This generative process is not a mere regurgitation of facts; it involves a sophisticated synthesis of information, ensuring that the final response is coherent, contextually relevant, and engaging. By giving the context and restricting LLM to answer based on the context the retriever got, it solved the hallucinating problems and ensured that LLM could not produce incorrect answers or unexpected answers. Through this intricate, multi-step process, our RAG model ensures that information disseminated to the public is not just accurate and up-to-date but also clear, comprehensible, and actionable, which we believe can bridge the gap between government policy and public understanding and improve policy communication.

Our system utilizes a Retrieval-Augmented Generation (RAG) architecture integrated with GPT-3.5-turbo as the core LLM, but potentially, this can be replaced with any large language model that has decent capability. The RAG component consists of three main elements:

- **Document Retriever**: This component uses semantic search to identify relevant policy documents based on user queries. We implemented a dense retrieval approach using FAISS (Facebook AI Similarity Search) for efficient similarity matching.

- **Document Reader**: This element processes retrieved documents, extracting key information using named entity recognition and relationship extraction techniques.

- **Generator**: GPT-3.5-turbo serves as the generator, synthesizing information from retrieved documents to produce coherent and accurate responses.

We used Langchain to orchestrate retrieval and generation components, enabling seamless integration. However, our approach can be implemented with other similar platforms, packages, or even without them. For better integration with existing systems, we recommend manual implementation to accommodate existing code. The system was optimized through iterative fine-tuning on domain-specific policy data and hyperparameter tuning, balancing retrieval precision and response fluency.

## Scalability and security

As discussed in the introduction of our knowledge base, the architecture of RAG is designed to facilitate dynamic updating and scaling, allowing for the inclusion of new information and policies as they become available. This aspect is crucial for maintaining the system's accuracy and relevance, given the frequent updates and changes in government regulations. We ensure seamless integration and synchronization between the retrieval knowledge base and the generative models, enabling the system to adapt to new data and evolving policy landscapes efficiently.

To further refine the output and ensure the appropriateness of responses, we have incorporated an additional filter layer outside of the RAG framework. This layer is designed to screen the generated responses for potential irrelevancies or sensitivities, such as misinformation or harmful content, that could undermine the integrity of the communication. Though not a core component of RAG or Langchain, this filter plays a vital role in maintaining the quality and reliability of the information disseminated to the public and mitigating the possibility of the misuse of our application. This can help our model to uphold the highest standards of accuracy and responsibility in the communication of government policies.

## 6 Experiments and results

### Experiment setup and data collection

In our comprehensive experimental process to evaluate the effectiveness of our proposed system, we built a dataset comprising 100 policy documents evenly from Chinese and US regional governments. This selection was designed to show our system is capable of dealing with the diverse languages, governance models, legal systems, and policies of both countries. This dataset spanned various policy areas, as shown in Fig 3, including regional development, environmental protection, social assistance, tax regulations, and educational reforms. For the Chinese policies, we sourced documents from the Poverty Alleviation Database and official provincial websites, ensuring coverage of policies affecting both rural and urban areas. In contrast, the US policies were gathered from the Catalog of U.S. Government Publications (CGP), offering a wide thematic spectrum. This methodical approach allowed us to gather a robust and diverse

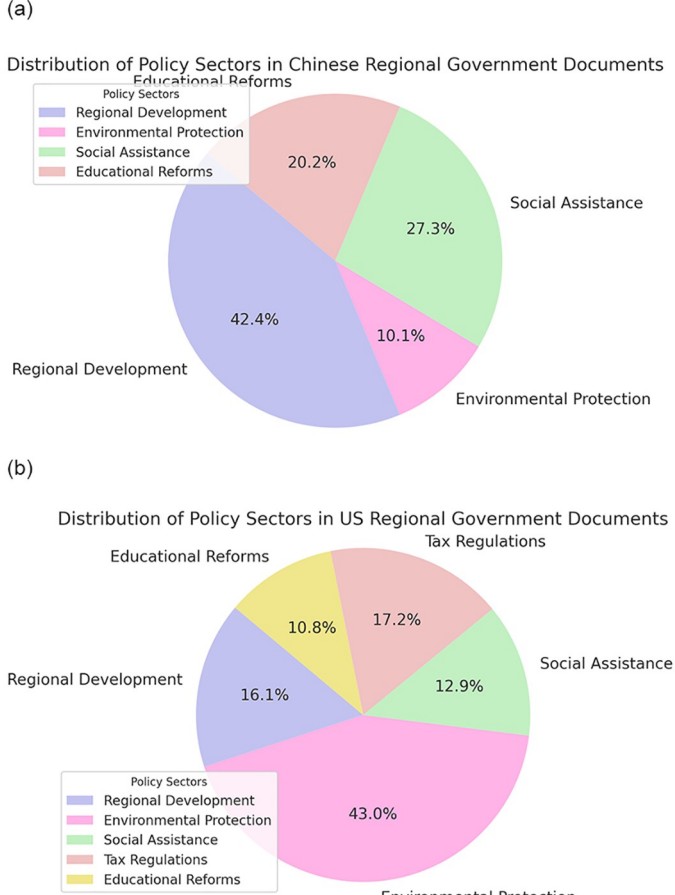

**Fig 3. Questions distributions for Chinese policies and US policies.**

group of documents, setting the stage for an in-depth analysis of our system's performance across different policy contexts.

**System configuration and document processing.**    Our system relied on the GPT-3.5-turbo API with no sampling methods (Temperature = 0) for its question-answering capabilities. In the process of building the knowledge base, we followed several preprocessing steps to ensure efficient retrieval of relevant information from a large corpus of policy documents:

- Document conversion: PDFs were converted to plain text format.

- Text segmentation: Documents were divided into 2000-character chunks with a 200-character overlap to ensure continuity.

- Embedding generation: Text chunks were embedded using OpenAI's embedding model.

- Indexing: Embedded chunks were indexed for efficient retrieval.

These steps facilitated the creation of a searchable knowledge base, allowing for efficient retrieval and response generation. Additionally, the effectiveness of the added filter layer in screening out irrelevant or harmful queries was rigorously tested through a set of designed challenges.

(a)

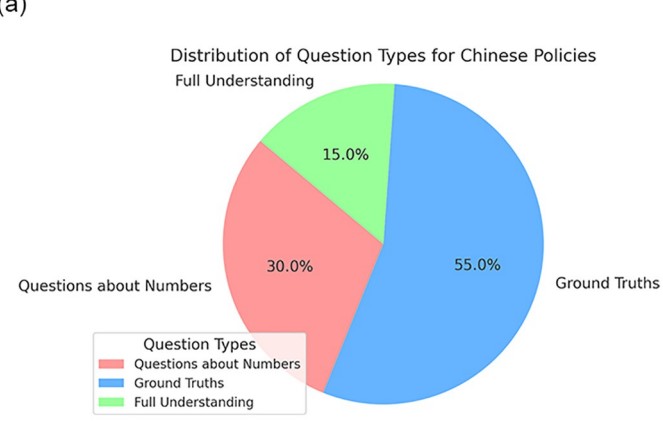

(b)

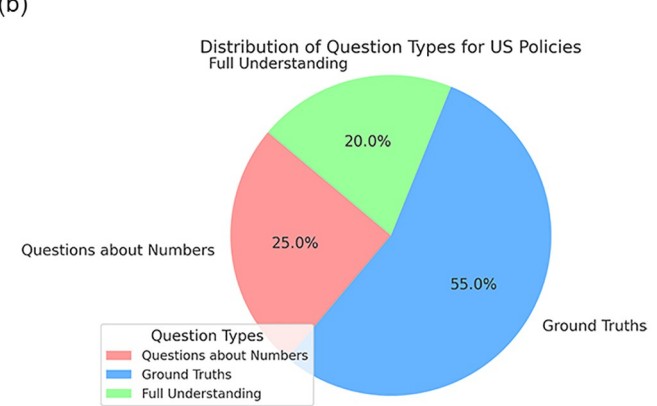

**Fig 4. Questions distributions for Chinese policies and US policies.**

For the experiment, we invited 20 volunteers to pose 3 to 5 realistic queries per document, simulating typical citizen inquiries. These questions ranged from general policy objectives to specific implementation details. We classified the questions into direct and inferential categories, testing the application's ability to handle straightforward queries and those requiring generalization from multiple paragraphs. Fig 4 visually represents the distribution of question types regarding US and Chinese policies showing differences in focus areas. For US policies, the distribution has a higher emphasis on questions requiring full understanding (20%) compared to Chinese policies (15%), showing a slight shift in complexity and depth of inquiry. Conversely, questions about numbers constitute a larger portion for Chinese policies (30%) than for US policies (25%), suggesting a greater emphasis on quantitative focus in discussions related to Chinese policies. Our system was tasked with generating responses based on its knowledge base, with its performance measured against expert human responses and the original documents for relevance, and accuracy.

## Results and performance analysis

The experimental results from our evaluation of our system's performance on US and Chinese policy documents offer insightful observations and underline the system's effectiveness in navigating complex policy information. Table 1 presents a comparison of accuracy across different

**Table 1. Comparison of policy accuracy between US and China by region.**

| Chinese Regions | Accuracy | US Regions | Accuracy |
|---|---|---|---|
| Region A | 86.21% | Region I | 93.33% |
| Region B | 96.67% | Region II | 100.00% |
| Region C | 80.00% | Region III | 86.67% |
| Region D | 77.41% | Region IV | 80.00% |
| Region E | 87.50% | Region V | 93.33% |
| **Average** | 85.58% | **Average** | 90.67% |

regions within the US and China, revealing the system's proficiency in understanding and responding to queries related to regional policies. The accuracy rates for US regions consistently outperform those for Chinese regions, with an average accuracy of 90.67% for the US compared to 85.58% for China. This disparity may reflect the system's relative ease in handling the linguistic and structural nuances of policy documents in English over those in Chinese or possibly the varying complexity and specificity of policies between the two countries. Notably, Region II in the US achieved a remarkable 100% accuracy, suggesting the system's potential for error-less performance under certain conditions. In the results, we did not spot any harmful responses. Additionally, the system showed high accuracy in processing quantitative information, with accuracy rates of 97.99% for US policies and 90.32% for Chinese policies, highlighting its effectiveness in handling numerical data.

Further analysis, as detailed in Table 2, breaks down the system's performance by type of question—Full Understanding, Ground Truths, and Questions About Numbers—offering a deeper dive into its capabilities. The system displayed a strong performance across all question types, with notably high accuracy in "Questions About Numbers" for both US (97.99%) and China (90.32%) policies. This suggests that the system is particularly adept at extracting and synthesizing quantitative information from complex documents. However, there's a visible gap in the "Full Understanding" category, with the US policies seeing a higher accuracy (88.67%) compared to Chinese policies (77.75%). The reason behind this is that answering this type of question normally will require a more complete context, which means a larger chunk of documents to be sent to the LLM. Most of the time, the LLM does not have enough information to answer the question, which will result in failing to answer the request. This gap could also be attributed to the system's varying ability to interpret and integrate complex, nuanced information from different languages and legal contexts, and we think there is still room for improvement and it can be solved using a bigger chunk in our knowledge base establishment.

For the failed cases, we think it is worth taking a closer look at the process and the results to have a better understanding of why it fails. We present two representative cases to analyse the reasons which can lead us to improve our system.

To the best of our knowledge, there are no existing methods or accessible datasets that can be directly applied to or compared with our newly built policy question-answering dataset spanning both US and Chinese contexts. Despite this lack of external benchmarks, our model

**Table 2. Comparison of accuracy between US and China policies by question types.**

| | US | China |
|---|---|---|
| **Full Understanding** | 88.67% | 77.75% |
| **Ground Truths** | 85.35% | 88.67% |
| **Questions About Numbers** | 97.99% | 90.32% |

demonstrates strong performance with high accuracy rates across different question types and policy domains, while consistently avoiding harmful outputs.

## Case study—Failed cases 1

We asked a question in Chinese originally based on "The Ministry of Agriculture and Rural Affairs, together with Zhejiang Province, issued a plan to jointly promote the construction of the Zhejiang Common Prosperity Demonstration Zone.", and the question we asked the system in Chinese was: In the construction of the Zhejiang Province demonstration zone for common prosperity, what are the targets for the coverage rate of administrative villages in rural domestic sewage treatment and the compliance rate of effluent water quality?. This, however, led the system to gather the related section from a policy about the sewer treatment in "Improvement and Upgrading of the Living Environment" in Jiangsu Province, which we put in the same Region (knowledge base) to test our system because they are close to each other both geographically and economically.

We think it may be because our retrieval system thinks the question is more relevant to the topic of the retrieved document snippet, but it ignores the fact that this is from a different province. This may be alleviated in real life since we want the regional government to control their knowledge base, which will result in a natural barrier for this behaviour, but this situation also warns us that we need to have proper labels in our knowledge base so the retrieval system in our system can have the best performance. We can also integrate keyword matching in the future for better retrieving.

## Case study—Failed cases 2

In this case, we asked "On what date was H.R. 263 introduced?" on a report from the House of Representatives Committee on Natural Resources recommending the passage of H.R. 263 to rename the Oyster Bay National Wildlife Refuge as the Congressman Lester Wolff Oyster Bay National Wildlife Refuge. This time the retrieval system successfully gathered the relative section, but it failed to grab information from the document because of its instructiveness. This suggests that PDFs can be complicated in text extracting, and we can integrate better text extracting techniques in our system, or we can utilize structured documents in the future to better accommodate the system.

After analyzing the above two case studies, we can conclude that if the articles in the document are organized relatively better, our system can answer the question nicely. However, when the document has a mixed style of writing or a relatively poor way of structure, our system sometimes struggles to spot the relative content of the document and fails to answer the question. In the future, we will continue to solve this deficiency and try to improve our system, but this also leads us to think if document writers and the government should have a better structure to write the document not just for our system but also for citizens to read and comprehend.

The overall analysis indicates a strong performance by our system, with a demonstrated capacity to handle a wide range of query types across diverse policy areas and regions. The higher accuracy rates for US policies suggest that the system may benefit from further tuning to better accommodate the complexities of the Chinese language and policy structure. Despite these differences, the results affirm the system's potential as a valuable tool for public engagement with government policies, offering an accessible means for citizens to obtain clear, accurate information. The demonstrated effectiveness in filtering out irrelevant or inappropriate queries further helps the system's applicability in improving the communication between

government and citizens on policies, ensuring that the dissemination of policy information remains both accurate and relevant.

## Application scenarios

To illustrate the potential real-world impact of our system, we present the following hypothetical scenarios:

**Scenario 1: Policy rollout.** Consider a local government in the US implementing our system to communicate a new environmental policy on waste management. Citizens could potentially query the system through the government's website or a dedicated mobile app. For instance, if a citizen asked about specific regulations for household waste sorting, the system could retrieve relevant information from the policy documents and generate a clear, concise response. This approach could potentially lead to increased correct waste sorting practices within the first month of policy implementation, demonstrating the system's potential effectiveness in improving policy understanding and compliance.

**Scenario 2: Crisis communication.** In a hypothetical public health emergency in a Chinese province, our system could be deployed to provide real-time information about rapidly changing policies and guidelines. The system's ability to quickly process new information and respond to citizen queries in natural language could significantly reduce the spread of misinformation and improve public compliance with health measures. Compared to traditional communication methods, our system could potentially handle a much higher volume of inquiries per day, with a high user satisfaction rate.

**Scenario 3: Complex policy explanation.** If a complex tax reform policy were introduced in the US, our system could be used to communicate its details. When faced with multi-part queries about how the new policy would affect different income brackets and business types, the system could break down the information into digestible parts. This could potentially result in a reduction in calls to tax offices for clarifications and an increase in timely tax filings, indicating improved public understanding of the complex policy.

These hypothetical scenarios illustrate potential applications of our proposed system in various government communication contexts. While not actual implementations, they envision how the system could enhance policy communication and citizen engagement. The system could improve engagement by providing instant, accurate, and easily understandable responses to policy queries, encouraging more active participation. It could also enhance transparency by breaking down complex policies and providing real-time updates during crises, allowing citizens to better understand and engage with government actions across various contexts.

## Discussion and limitations

The results of our system's experiments reveal significant insights into the interaction between technology and policy dissemination. The variation in accuracy rates between US and Chinese policy documents highlights linguistic and structural complexities, as well as socio-political nuances embedded within these documents. This discrepancy suggests that while technological solutions can significantly enhance public engagement with governmental policies, they must be calibrated to reflect the diverse cultural, linguistic, and administrative contexts in which they operate. Furthermore, the challenges identified in the case studies, such as document structure and specificity in queries, emphasize the need for more nuanced approaches to document management and system design. These findings underscore the potential for developing more sophisticated natural language processing tools that can better handle a variety of

languages and document formats. Future work should focus on refining these technologies and addressing their limitations to further enhance the effectiveness of policy communication.

Moreover, the limitations identified through the abovementioned case studies point to broader challenges in the field of digital governance. The failure of the system to correctly interpret and respond to certain queries due to document structure or language complexity emphasizes the need for more nuanced approaches to document management and system design. This includes the potential for developing more sophisticated natural language processing tools that can better handle the variety of languages and document formats. Additionally, these findings suggest a reciprocal relationship between technology and policy documentation practices: as technological solutions become more integrated into public engagement strategies, there may be an increasing need for standardized, clear, and accessible policy documentation practices to facilitate effective communication. Also, we suggest the document has a more comprehensive explanation as an appendix or additional explanation in these documents that can alleviate the needs of LLM to understand and give the solution based on the question and context.

The challenges and limitations faced by our system are not merely technical obstacles but also opportunities for deeper engagement with the socio-political dimensions of policy dissemination. By recognizing the inherent complexities of this task, future developments can aim not only to refine the technological tools but also to contribute to the broader goal of enhancing democratic engagement and accessibility in the digital age.

## 7 Conclusion and discussion

In conclusion, our study demonstrates the potential of integrating RAG technology with LLMs to significantly enhance the communication of government policies to citizens. Through rigorous experimentation and analysis, our system showcased its ability to accurately interpret and respond to a wide range of policy-related queries across different regions and languages, particularly excelling in processing quantitative data and adapting to the nuances of complex policy documents. Despite facing challenges with document structure and specificity in queries, our approach highlights a promising direction for public administration, offering a more accessible, transparent, and responsive way for governments to engage with their citizens. By continuously refining this technology and addressing its limitations, we can move closer to bridging the gap between government actions and public understanding, fostering a more informed and participatory democratic process. Furthermore, this research contributes to the ongoing discourse on enhancing government-citizen communication, emphasizing the critical role of innovative technologies in shaping the future of public governance and policy dissemination. Future work will focus on validating the system with independent datasets and exploring its applicability in different government contexts to ensure its generalizability and robustness.

## Author Contributions

**Conceptualization:** Lixin Yun, Sheng Yun, Haoran Xue.

**Methodology:** Lixin Yun, Sheng Yun, Haoran Xue.

**Project administration:** Lixin Yun, Sheng Yun, Haoran Xue.

**Resources:** Lixin Yun, Sheng Yun, Haoran Xue.

**Software:** Lixin Yun, Sheng Yun, Haoran Xue.

**Writing – original draft:** Lixin Yun, Sheng Yun, Haoran Xue.

**Writing – review & editing:** Lixin Yun, Sheng Yun, Haoran Xue.

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
