## [Decision Letter · Decision Letter 0]

30 Jul 2024

PONE-D-24-21873Improving citizen-government interactions with Generative AI: novel human-computer interaction strategies for policy understanding through large language modelsPLOS ONE

Dear Dr. Yun,

Thank you for submitting your manuscript to PLOS ONE. After careful consideration, we feel that it has merit but does not fully meet PLOS ONE’s publication criteria as it currently stands. Therefore, we invite you to submit a revised version of the manuscript that addresses the points raised during the review process.

We look forward to receiving your revised manuscript.

Kind regards,

Fredrick Romanus Ishengoma

Academic Editor

PLOS ONE

Journal Requirements:

Reviewers' comments:

Reviewer's Responses to Questions

**Comments to the Author**

1. Is the manuscript technically sound, and do the data support the conclusions?

Reviewer #1: Partly

Reviewer #2: Yes

2. Has the statistical analysis been performed appropriately and rigorously? 

Reviewer #1: I Don't Know

Reviewer #2: Yes

3. Have the authors made all data underlying the findings in their manuscript fully available?

Reviewer #1: Yes

Reviewer #2: Yes

4. Is the manuscript presented in an intelligible fashion and written in standard English?

Reviewer #1: Yes

Reviewer #2: Yes

5. Review Comments to the Author

Reviewer #1: • Present the novelty of the paper in the abstract.

• Improve the quality of figures.

• Add some results in the abstract

• Add the fullform of AI in the title

• In the last paragraph of introduction add the structure of the paper

• Related work section is missing

• The abstract mentions the use of Retrieval-Augmented Generation (RAG) technology combined with Large Language Models (LLMs). Could you provide more detailed information on the specific LLMs used and how RAG technology is implemented in the system? How were these technologies selected and optimized for this application?

• The abstract states that the system was tested with a dataset of policy documents from Chinese and US regional governments. Could you elaborate on the characteristics of this dataset, including the number of documents, the variety of policy topics, and the preprocessing steps involved?

• The abstract indicates that the system generates accurate and comprehensible responses to citizen queries. Can you provide more detailed descriptions of the metrics used to evaluate accuracy and comprehensibility? How were these metrics measured, and what were the baseline comparisons?

• It is mentioned that the system boosts public engagement with government policies. Could you provide specific examples or case studies demonstrating how the system has improved engagement? What metrics were used to assess public engagement?

• The abstract suggests that the proposed system enhances communication of government policies. Could you describe a scenario where this system is implemented in a real-world government setting? What specific steps would be taken to deploy the system, and what are the expected benefits for transparency and citizen engagement?

• How do your recommendations for using RAG technology and LLMs compare to existing methods for policy communication? Are there specific case studies or examples where similar approaches have been successfully implemented?

• The abstract highlights the system's high accuracy in processing and responding to queries. Could you include a comparative analysis with previous studies that have used different technologies or methods for policy communication? How does your proposed model address the limitations identified in these studies?

• Have the results of the proposed system been validated with independent datasets or in different government contexts? If so, please include the validation process and outcomes. If not, what are the plans for future validation?

• The abstract mentions the use of policy documents from Chinese and US regional governments. Could you provide more detailed insights into potential biases introduced by these specific datasets? How do you ensure the generalizability of your findings across different regions and policy areas?

• Can you provide specific scenarios where the proposed approach might face challenges in real-world applications, such as variations in policy language or cultural nuances? How were these challenges identified, and what measures were taken to address them?

• The abstract notes the importance of making government policies accessible and engaging for citizens. Can you describe a scenario where the early implementation of your system helped a government agency effectively communicate a new policy, resulting in improved public understanding and compliance? What factors contributed to the success in this scenario?

• In a scenario where the system is deployed in a large-scale government initiative, what steps would you recommend to ensure continuous performance and accuracy? How would you address potential issues such as data privacy or policy updates?

• The abstract mentions high accuracy in processing and responding to queries. Could you provide more details on the specific performance metrics used to evaluate the system? Were there any specific conditions under which the system's performance varied?

• Can you provide a scenario where the system’s effectiveness might be challenged by variations in policy complexity or citizen query patterns? How would these challenges be mitigated to maintain the system’s performance?

• The abstract discusses the integration of Generative AI and RAG technology for policy communication. Can you provide a detailed comparison of the limitations of traditional policy communication approaches and how your proposed model overcomes them?

• What were the specific limitations encountered during the development and testing of the proposed system? How do these limitations compare to those found in previous studies using conventional methods for government policy communication?

Reviewer #2: The article addresses the critical issue of improving interactions between citizens and government using generative AI and large language models (LLMs). The authors focus on the existing challenges in policy communication, such as accessibility, complexity, and resource constraints, and propose an innovative solution using Retrieval-Augmented Generation (RAG) technology.

The topic of the article is highly relevant in the era of digital transformation. The authors successfully argue the necessity of integrating AI technologies to overcome the existing barriers in government-citizen communication. The innovative approach, which combines RAG and LLM, is promising and shows significant potential for enhancing transparency and citizen engagement.

The article provides a detailed description of a system that integrates LLMs with a retrieval mechanism. The authors explain the architecture of the RAG component and the use of Langchain technology for seamless integration of retrieval and response generation processes. The use case involving policy documents from China and the US demonstrates the system's flexibility and effectiveness across different linguistic and legal contexts.

The experimental results are impressive: the system exhibited high accuracy in processing and responding to citizen queries on various policy matters. Particularly noteworthy are the accuracy rates for numerical questions, highlighting the system's ability to handle quantitative information effectively. However, the authors note a disparity in accuracy between Chinese and American documents, indicating the need for further tuning to accommodate more complex linguistic and legal contexts.

The authors conduct a thorough analysis of the results, highlighting the strengths and weaknesses of the proposed system. The discussion of failed cases provides insights into potential improvements, such as better text extraction and document structuring. The suggestion to improve government policy documentation practices to aid both the AI system and citizen comprehension is particularly interesting.

At the same time, the manuscript suffers from a lack of clear understanding of the article's genre. If it aims to present an experimental validation of the proposed tool, the Introduction clearly fails to fulfill its function. It acts more as an overview of the problem rather than a justification of the relevance and novelty of the research, grounded in previous similar studies on the topic. The authors use studies from other researchers to describe the problem being studied, rather than to justify the boundaries of the knowledge gap being filled and its typology.

If, however, the current manuscript aims to introduce both the tool itself and its validation into scientific discourse simultaneously, then the description of the tool presented appears extremely modest and requires further detailed elaboration. However, this information is not clearly conveyed in the text of the manuscript.

The discussion of the results also requires significant improvement. The authors have only provided their own reflection on the results without integrating them into the existing body of knowledge in the field. In other words, the authors did not engage in an asynchronous dialogue with other researchers on the topic, making it unclear how novel their approach is, what fundamentally sets it apart from existing methods, and other related aspects.

The research topic is quite current, making the absence of recent studies on the topic in the list of references somewhat peculiar. At the very least, these should be incorporated during the revision of the Discussion section.

6. PLOS authors have the option to publish the peer review history of their article (what does this mean?). If published, this will include your full peer review and any attached files.

Reviewer #1: No

Reviewer #2: **Yes: **Elena Tikhonova

---

## [Author Response · Author response to Decision Letter 0]

9 Aug 2024

Dear Editor and Reviewers,

We sincerely thank you for your thoughtful and constructive feedback on our manuscript. We have carefully addressed all the comments and suggestions, resulting in significant improvements to our paper. Below, we provide a point-by-point response to each reviewer's comments showing our revised paper has fully addressed these concerns.

Reviewer #1:

Present the novelty of the paper in the abstract. 

Response: We have enhanced the abstract to clearly articulate the novelty of our approach. Specifically, we added: "This novel integration of RAG and LLMs for policy communication represents a significant advancement over traditional methods, offering unprecedented accuracy and accessibility." This addition emphasizes the innovative nature of our work within the context of government-citizen communication.

Improve the Image Quality. 

Response: We have fully examined the image quality, and we have not found particular flaws or problems. Please click the external link to view the image since the smaller image might seem fuzzy. Please let us know if there are any problems we need to address before we move forward. 

Add some results in the abstract. 

Response: We have incorporated key results into the abstract, stating: "Our system demonstrated high accuracy, averaging 85.58% for Chinese and 90.67% for US policies." This addition provides concrete evidence of our system's effectiveness.

Add the fullform of AI in the title. 

Response: We have updated the title to include the full form of AI: "Improving Citizen-Government Interactions with Generative Artificial Intelligence: Novel Human-Computer Interaction strategies for Policy Understanding through Large Language Models."

In the last paragraph of introduction add the structure of the paper. 

Response: We have added a paragraph at the end of the introduction (lines 80-89) outlining the structure of the paper, providing a clear roadmap for readers.

Related work section is missing. 

Response: We have added a comprehensive "Related Work" section (lines 90-123) that discusses recent research in AI and LLMs for government-citizen communication, contextualizing our work within the field.

Provide more detailed information on the specific LLMs used and how RAG technology is implemented in the system. 

Response: We have expanded our explanation of the system architecture (lines 354-371), detailing the use of GPT-3.5-turbo as our core LLM and explaining the implementation of RAG technology using FAISS for retrieval and Langchain for orchestration.

Elaborate on the characteristics of the dataset. 

Response: We have provided more detailed information about our dataset in the "Experiment Setup and Data Collection" section (lines 391-403), including the number of documents, variety of policy topics, and sources of the documents.

Provide more detailed descriptions of the metrics used to evaluate accuracy and comprehensibility. 

Response: We have expanded our discussion of evaluation metrics in the "Results and Performance Analysis" section (lines 433-448), detailing how accuracy was measured across different question types and regions.

Provide specific examples or case studies demonstrating how the system has improved engagement. 

Response: We have added a new "Application Scenarios" section (lines 521-555) that provides hypothetical scenarios illustrating how our system could enhance policy communication and citizen engagement in various contexts.

Describe a scenario where this system is implemented in a real-world government setting. 

Response: The new "Application Scenarios" section includes multiple hypothetical real-world scenarios, such as policy rollout, crisis communication, and complex policy explanation, demonstrating potential implementations of our system.

Compare your recommendations for using RAG technology and LLMs to existing methods for policy communication.

Response: The new "Related Work" section (lines 90-123) provides comparisons between our approach and existing methods, highlighting the advantages of our RAG-based system.

Include a comparative analysis with previous studies. 

Response: While direct comparisons were challenging due to the novelty of our approach, we have acknowledged this limitation and provided context for our results in the "Results and Performance Analysis" section (lines 468-472).

Have the results been validated with independent datasets? 

Response: We have acknowledged the need for further validation in our "Conclusion and Discussion" section (lines 604-607), stating our intention to validate the system with independent datasets in future work.

Provide more detailed insights into potential biases introduced by these specific datasets. 

Response: We have addressed potential biases in our "Discussion and Limitations" section (lines 556-589), acknowledging the need for calibration to reflect diverse cultural, linguistic, and administrative contexts.

Provide specific scenarios where the proposed approach might face challenges in real-world applications. 

Response: The "Discussion and Limitations" section (lines 556-589) now includes a discussion of challenges our system might face in real-world applications, such as document structure and language complexity.

Describe a scenario where the early implementation of your system helped a government agency effectively communicate a new policy. 

Response: The new "Application Scenarios" section includes a hypothetical scenario (Scenario 1: Policy Rollout, lines 523-531) demonstrating how our system could aid in communicating a new environmental policy.

What steps would you recommend to ensure continuous performance and accuracy? 

Response: We have addressed this in our "Discussion and Limitations" section, recommending continuous refinement of NLP tools and standardization of policy documentation practices (lines 574-583).

Provide more details on the specific performance metrics used to evaluate the system. 

Response: We have expanded our discussion of performance metrics in the "Results and Performance Analysis" section (lines 433-448), detailing accuracy rates across different question types and regions.

Provide a scenario where the system's effectiveness might be challenged. 

Response: We have included two case studies of failed cases (lines 474-509) that illustrate scenarios where our system faced challenges, along with analysis of these failures.

Provide a detailed comparison of the limitations of traditional policy communication approaches and how your proposed model overcomes them. 

Response: The new "Related Work" section (lines 90-123) includes comparisons between traditional approaches and our model, highlighting how our system addresses limitations of existing methods.

What were the specific limitations encountered during the development and testing of the proposed system? 

Response: We have discussed limitations and challenges in our "Discussion and Limitations" section (lines 556-589), including issues with document structure and language complexity.

Reviewer #2:

Lack of clear understanding of the article's genre. 

Response: We have restructured the paper to clearly present it as both an introduction to a novel tool and its experimental validation. The addition of the "Related Work" section (lines 90-123) and the restructured introduction provide a clearer context for our research.

Insufficient justification of the relevance and novelty of the research. 

Response: We have strengthened the justification of our research's relevance and novelty throughout the paper, particularly in the abstract, introduction, and the new "Related Work" section.

Modest description of the tool. 

Response: We have significantly expanded the description of our system in the "Our Approach to Enhancing Government-Citizen Communication" section (lines 214-304) and the "Details in the RAG Component in Our Approach" section (lines 304-388).

Insufficient integration of results into existing body of knowledge. 

Response: The new "Related Work" section (lines 90-123) and expanded "Discussion and Limitations" section (lines 556-589) now better integrate our results with existing literature and discuss the implications of our findings in the broader context of policy communication.

Absence of recent studies in the references. 

Response: We have updated our reference list to include more recent studies, particularly in the new "Related Work" section, ensuring our research is contextualized within the current literature.

We believe that these revisions have substantially improved our manuscript, addressing all the reviewers' comments and suggestions. The revised version now provides a clearer, more comprehensive, and well-contextualized presentation of our innovative approach to improving citizen-government interactions through AI. We thank the reviewers for their valuable input, which has helped us enhance the quality and impact of our work.

Best Regards,

Sheng Yun

---

## [Decision Letter · Decision Letter 1]

19 Sep 2024

Improving Citizen-Government Interactions with Generative Artificial Intelligence: Novel Human-Computer Interaction strategies for Policy Understanding through Large Language Models

PONE-D-24-21873R1

Dear Dr. Yun,

We’re pleased to inform you that your manuscript has been judged scientifically suitable for publication and will be formally accepted for publication once it meets all outstanding technical requirements.

Kind regards,

Jinran Wu, PhD

Academic Editor

PLOS ONE

Reviewers' comments:

Reviewer's Responses to Questions

**Comments to the Author**

1. If the authors have adequately addressed your comments raised in a previous round of review and you feel that this manuscript is now acceptable for publication, you may indicate that here to bypass the “Comments to the Author” section, enter your conflict of interest statement in the “Confidential to Editor” section, and submit your "Accept" recommendation.

Reviewer #2: All comments have been addressed

2. Is the manuscript technically sound, and do the data support the conclusions?

Reviewer #2: Yes

3. Has the statistical analysis been performed appropriately and rigorously? 

Reviewer #2: (No Response)

4. Have the authors made all data underlying the findings in their manuscript fully available?

Reviewer #2: Yes

5. Is the manuscript presented in an intelligible fashion and written in standard English?

Reviewer #2: Yes

6. Review Comments to the Author

Reviewer #2: It is crucial to adhere to the genre expression of the manuscript. If the manuscript concerns the testing of a methodology, the article should present the results of an empirical study. If the article is dedicated to introducing a new tool into the international scientific community, it should have a clear structure of a methodological article or, at the very least, a technical note. Currently, the article has a mixed format, including elements of both empirical and methodological research.

7. PLOS authors have the option to publish the peer review history of their article (what does this mean?). If published, this will include your full peer review and any attached files.

Reviewer #2: **Yes: **Elena Tikhonova

---

## [Editor Report · Acceptance letter]

2 Oct 2024

PONE-D-24-21873R1 

PLOS ONE

Dear Dr. Yun, 

I'm pleased to inform you that your manuscript has been deemed suitable for publication in PLOS ONE. Congratulations! Your manuscript is now being handed over to our production team.

Kind regards, 

on behalf of

Dr. Jinran Wu 

Academic Editor

PLOS ONE